# Facilitating Communication in Neuromuscular Diseases: An Adaptive Approach with Fuzzy Logic and Machine Learning in Augmentative and Alternative Communication Systems

Jhon Fernando Sánchez-Álvarez [1], Gloria Patricia Jaramillo-Álvarez [2] and Jovani Alberto Jiménez-Builes [2,*]

[1] School of Basic Sciences, Technology and Engineering, Universidad Nacional Abierta y a Distancia, Bogotá 111321, Colombia; jhonf.sanchez@unad.edu.co
[2] Department of Computer and Decision Sciences, Faculty of Mines, Universidad Nacional de Colombia, Medellín 050034, Colombia; gpjarami@unal.edu.co
[*] Correspondence: jajimen1@unal.edu.co

**Abstract:** Augmentative and alternative communication techniques (AAC) are essential to assist individuals facing communication difficulties. (1) Background: It is acknowledged that dynamic solutions that adjust to the changing needs of patients are necessary in the context of neuromuscular diseases. (2) Methods: In order address this concern, a differential approach was suggested that entailed the prior identification of the disease state. This approach employs fuzzy logic to ascertain the disease stage by analyzing intuitive patterns; it is contrasted with two intelligent systems. (3) Results: The results indicate that the AAC system's adaptability enhances with the progression of the disease's phases, thereby ensuring its utility throughout the lifespan of the individual. Although the adaptive AAC system exhibits signs of improvement, an expanded assessment involving a greater number of patients is required. (4) Conclusions: Qualitative assessments of comparative studies shed light on the difficulties associated with enhancing accuracy and adaptability. This research highlights the significance of investigating the use of fuzzy logic or artificial intelligence methods in order to solve the issue of symptom variability in disease staging.

**Keywords:** augmentative and alternative communication; computer–human interaction; fuzzy logic; machine learning; adaptive systems



## 1. Introduction

Communication theory presents four basic elements: source, sender, message, and receiver [1]. Typically, the sender emits the message, which can be transmitted orally or textually, among other ways, and the receiver receives the message. This process is particularly complex in individuals with conditions such as amyotrophic lateral sclerosis (ALS). For instance, some studies indicate that 50% of patients die 18 months after diagnosis. Additionally, they lose the ability to communicate by the fourth month post-diagnosis. This loss of communication can lead to symptoms unrelated to ALS, such as depression or dementia. Augmentative and Alternative Communication (AAC) is used to facilitate communication in these cases, enabling patients to express desires, thoughts, and ideas with their surroundings [2].

Recent studies show that up to 1% of the global population experiences some degree of speech, language, or communication needs [3]. The loss of speech associated with severe paralysis and other medical complications has long been a barrier between patients and the outside world. AAC encompasses various processes that enhance, complement, or replace speech for individuals with complex communication needs [4].

In recent years, AAC systems have proven to be invaluable tools for enhancing the quality of life for individuals with verbal communication difficulties and, in general, "artificial intelligence tools have the capacity to transform AAC systems" [5]. These systems

include a wide range of technologies and strategies that enable individuals with speech disabilities, such as neuromuscular diseases, language disorders, or traumatic injuries, to effectively express their thoughts, emotions, and needs. As an example, the research paper [6] introduces a virtual assistant that operates via a brain–computer interface (BCI), thereby enhancing the interaction between the user (hands-free technology and virtual assistants in smartphone application management) and the machine. As technology advances and our understanding of user needs evolves, significant progress has been made in designing and developing AAC systems.

Although significant advances have been achieved in this domain, significant challenges remain. As discussed in Ju et al.'s systematic review, some researchers have proposed prospective studies in AAC systems with the aim of enhancing communicative interaction among speech-capable individuals. The review mentioned above [7] concludes that it is crucial to develop systems that not only possess intuitive and straightforward implementation, but also integrate efficient communication strategies. It also emphasizes the importance of actively collaborating with speech-language pathology specialists, with a special focus on the design of AAC systems. The ability of AAC systems to adjust as degenerative neuromuscular illnesses develop is one of the biggest concerns. People with these conditions experience changes in their communication abilities as the disease progresses, requiring solutions that dynamically adjust to their ever-changing needs.

Previous research has addressed this challenge from various perspectives. Some approaches have focused on predicting disease progression and periodically manually adjusting AAC systems [8,9]. Others have explored machine-learning techniques to adapt systems based on user-collected data. For instance, the VocalID project aimed to create personalized voices for individuals using AAC devices by combining the unique vocal characteristics of the target person with input from voice donors, resulting in the generation of a unique synthetic voice. The 2014 study by Mills et al. highlights the relevance and transformative potential of this approach in the context of AAC [10]. Although these methods have showed potential, there are still issues with accuracy, simplicity, and response time [5].

In this study, we introduce a novel and exciting perspective to address adaptability in AAC systems. Our approach is based on fuzzy logic, a technique that simulates uncertainty and imprecision in data. We propose a system that identifies the disease state through heuristic rules, allowing us to provide a unique and adaptable solution that optimally adjusts throughout the disease progression and the user's lifetime. Our aim is to optimize the performance of virtual keyboards used in AAC systems to enhance communication efficiency for users with disabilities. We will explore how fuzzy logic adapts to the context of neuromuscular diseases, where disease stages are difficult to precisely classify due to symptom variability over time in each individual. Within the discussion section, a comparison is made between the proposed approach and two studies that have been chosen from the literature and make use of virtual assistants in AAC systems for messaging and social networking purposes.

Our assumptions are based on the idea that the design and interaction provided to the user will be contingent upon the state of the disease. Consequently, by dynamically adapting to the condition or progression of the user's disease, AAC systems can experience substantial enhancements. During the process of revising existing AAC systems, we proposed that they be optimized through early identification of the disease's stage, thereby permitting customization to the particular attributes of the user. This adaptability would hold the potential to effectively facilitate the target user's communication. This article also highlights the importance of AAC systems in the context of neuromuscular diseases and how these systems can significantly improve patients' quality of life by providing an effective means of communication. By analyzing different neuromuscular diseases, it demonstrates how these conditions affect communication and presents some compensatory strategies used to enhance it.

The current study is specifically concerned with the lack of adaptability exhibited by AAC systems that are predefined for users. We are in search of systems that are more adaptable and capable of adjusting to changes as a disease develops. Hence, we suggest the implementation of fuzzy logic techniques for the early detection of the patient's disease stage (initial, middle, or advanced). This would enable the system to adjust in response to the unique circumstances of the patient.

It is important to mention and clarify that advanced machine learning techniques have been used in AAC systems, with promising results in terms of direct prediction and improvement of interfaces [5]. In contrast, our approach does not emphasize intricate techniques for processing patterns or images. Instead, we present a proposal that stands out for the application of fuzzy logic methods, aiming to identify the stage or state of the patient's illness (initial, middle, or advanced) at an early stage. This allows for the adaptation of the system in a specific manner to the individual's situation.

This article is structured as follows: In Section 2, the materials, methodologies, and associated research are detailed. The results are presented in Section 3. The discussion and prospective directions of research are elaborated in Section 4, while the conclusions are presented in detail in Section 5.

## 2. Materials and Methods

Neuromuscular diseases encompass a range of conditions characterized by muscle loss of control and atrophy. Notable among these conditions are amyotrophic lateral sclerosis (ALS), Duchenne muscular dystrophy, myotonic muscular dystrophy, and spinal muscular atrophy [2,11–13].

For example, Duchenne muscular dystrophy, an X-linked degenerative disease, arises due to the absence of the dystrophin protein, critical for muscle fiber stability and protection. This condition impacts speech and respiration, often requiring compensatory approaches, voice amplification, and AAC systems as communication becomes more restricted [14–16].

On the other hand, myotonic muscular dystrophy, an autosomal dominant disease, can manifest at birth or later stages. Myotonia, a characteristic symptom, refers to slow muscle relaxation after voluntary contraction. As it progresses, this disease can impact communication due to muscle weakness and reduced motor control [17–19].

Similarly, spinal muscular atrophy is a recessive hereditary disease involving the degeneration of motor neurons in the spinal cord, resulting in progressive muscle weakness. Severity varies considerably, and different phenotypes are defined based on heterogeneous clinical features. SMA can impact communication due to muscle weakness, poor head control, and respiratory impairment [20,21].

In the literature, research has explored the development of adaptive AAC systems to enhance communication in individuals with neuromuscular diseases. Some relevant works focus on adaptable keywords and design multimodal systems that allow choosing between different interaction methods, such as buttons, head movement, and eye tracking. However, the challenge of determining the most suitable hardware combination for each individual persists [11,16,22–24].

Compared to physical keyboards, virtual keyboards exhibit lower efficiency in typing, even for users without disabilities. The lack of tactile feedback and the reduced size of virtual keys negatively affect performance. This reduction is more noticeable in people with disabilities due to motor restrictions. Techniques such as keyboard and letter sequence design, as well as complex text predictions, have been applied to enhance efficiency. However, virtual keyboards still prove to be less efficient than physical ones [19,25,26].

Fuzzy logic emerges as a promising tool to tackle the complexity and uncertainty of neuromuscular diseases. Given the diversity in symptoms and progression of these conditions, traditional classification techniques may be insufficient. Fuzzy logic addresses the imprecision of these diseases by representing concepts like "initial", "middle" and "advanced" stages through functions that reflect uncertainty. It provides accurate and adaptable stage classification based on the knowledge of neurological experts. Furthermore, fuzzy logic

is applicable in designing adaptive AAC systems. By considering motor skills and preferences, it determines optimal and personalized systems, thereby enhancing communication and interaction, contributing to patients' quality of life and autonomy [16,22,23].

### 2.1. Related Works

Machine learning (ML) has emerged as a revolutionary field in different areas. In this study, we emphasize the potential for transforming personalized adaptation in AAC systems. These systems, originally designed to assist individuals with communication disabilities in communicating, have experienced significant progress [5] as a result of the implementation of ML. By enabling AAC systems to dynamically adapt to the unique requirements and capabilities of users, this strategy has not only brought about significant efficiency improvements, but also enhanced the communicative experience.

Our study focused on two tools that leverage virtual assistants for social networking and messaging in pre-existing AAC systems designed for individuals with motor disabilities. We meticulously emphasized the ways in which these tools converge to offer innovative solutions that empower those who encounter diverse limitations in their communication capabilities.

Currently, virtual assistance tools use intelligent systems and ML, for example, in the study by [6], a BCI designed to restore communication skills in patients with severe motor disabilities is described. The BCI system controls four messaging applications. "Control of the BCI is achieved through the well-known visual P300 row-column (RCP) paradigm, which allows the user to select control commands and typing characters" [6]. In that study, they evaluated the system based on software usability performance standards, with experts and subjective surveys to healthy individuals who used the system. The process begins when the user sends a synthetic voice command generated by the BCI system and is recognized by the smartphone's virtual assistant. This generates the use of the messaging application and, consequently, communication with another user. It is important to note that BCI systems when evaluated in people with some kind of disability often have difficulties to achieve sufficient accuracy when using visual P300 CPR paradigms as mentioned in studies [27,28]; these studies did not find a clear correlation between the degree of disability and BCI performance [6]. The results obtained in this study, assessed through the implementation of usability questionnaires utilizing the System Usability Scale (SUS) [29], yielded positive scores for the tool, with an overall usability rating of 82.5. This score exceeds the threshold of 70, which is considered as a reliable indicator of an optimal level of usability for patients. Individual scores on subdimensions were successful, indicating that the application was easy to use and intuitive. Regarding the Raw NASA-TLX questionnaire [30], the average workload was 31.55, which is considered reasonably low. The subdimensions showed similar results to previous studies. The ad hoc questionnaire items provided indications of areas for improvement, such as the interface aesthetics. Overall, the subjective questionnaires suggest that the system was easy to control and pleasant for most participants, but they can be used to improve the system. We recommend evaluating the application's utility within this particular context through testing it on patients who have motor disabilities.

Furthermore, "different interpretations of the same input" is referred to in the study, which implies the fact that the way in which an individual speaks (the generation of the synthetic voice) can be understood differently depending on the context in which it occurs. As for those such as "time limit for replying to an incoming message, use of a similar name for different contacts and confusion between the subject and the body of the message"; related to the interaction of the system with the virtual assistant, they refer to specific problems when using voice commands to control the smartphone. In some cases, the assistant had difficulty understanding and requested more information from the user, but since only predefined commands were available, appropriate responses could not be provided, preventing the task from being completed [6].

The second chosen tool was developed by [31]. This research paper describes a specialized system that employs a BCI system to enable users to control social networks on a smartphone. Unlike the previous study, this research included assessments of patients with motor disabilities of some nature. By developing a system that captures signals from the brain, individuals are now capable of transmitting commands to applications such as Twitter and Telegram without the need to physically move. Throughout the assessment, various components of the brain control system were evaluated. It was evaluated with the participation of 10 healthy volunteers and 18 individuals with motor disabilities. Results demonstrated that the system achieved an accuracy rate of 92.3% among individuals in good health and 80.6% among those with motor impairments. The proposal may prove advantageous in homes, rehabilitation centers, and businesses, enhancing the lives of individuals through the offering of greater autonomy and independence. The aforementioned assessment utilized both quantitative and qualitative metrics; for the former, the time taken to complete each task, the number of correct selections, errors, and sequences were documented. In addition to calculating accuracy and output characters per minute (OCM), users were requested to fill out a questionnaire regarding the latter at the end of the session. The questionnaire comprised 20 items that were to be evaluated using a seven-point Likert scale [32]. Subjective opinions were collected regarding, among other things, session duration, user motivation, expectations, and application speed. An open-ended question was incorporated in order to gather recommendations [31]. It is crucial to note that the researchers employed the aforementioned qualitative evaluation technique in our study through the careful selection of ad hoc items.

## 2.2. Methodology

To conduct this study, a four-step methodology was implemented, as shown in Figure 1. In the initial stage (construction stage), three representative participants with different neuromuscular disease conditions were selected: individuals diagnosed with amyotrophic lateral sclerosis (ALS), muscular dystrophy, and spinal cord injury. In stage 2, we proceeded to the extraction and rigorous identification of the symptoms of each disease in collaboration with expert neurologists specialized in movement disorders (see Figure 1).

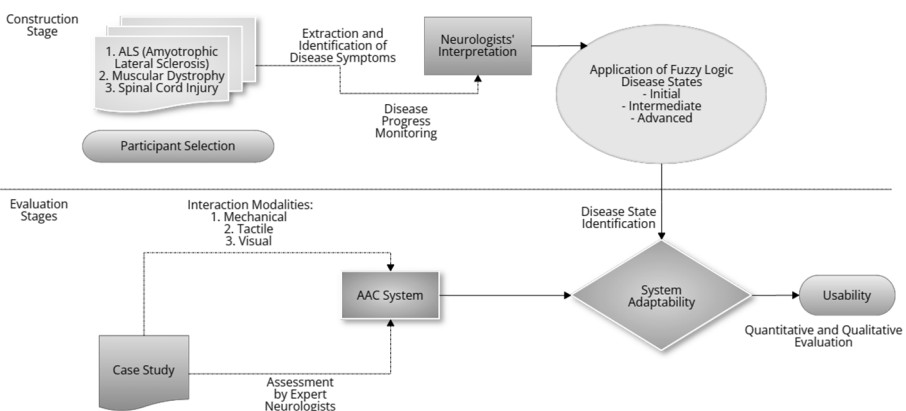

**Figure 1.** Methodology proposed. Source: Authors.

Following this, in the third stage of the system's development, a fuzzy logic approach was implemented to classify the disease states as initial, middle, and advanced. Throughout the assessment phases, three distinct modes of interaction were employed: visual, tactile, and mechanical. An adaptive AAC system was implemented, which was designed specifically to dynamically adapt to the evolving motor skills and communication preferences of the participants as the disease progressed.

A longitudinal design was employed for each case study, wherein the adaptability of the AAC system was assessed in both simulated and real-life scenarios. The precise diagnosis of the disease state and the determination of the system's efficacy were both

significantly influenced by the evaluation of proficient neurologists. Finally, in stage fourth, a quantitative and qualitative evaluation was performed, recording time metrics, errors and using specific questionnaires to assess the efficiency, effectiveness, ease of learning, satisfaction, and overall usability of the system.

*2.3. Experimental Design*

Three fundamental phases comprise the design for assessing the implementation of the AAC system in patients with neuromuscular disorders. Initial phase: calibration, during which the system is tailored to each participant by collecting data and administering specialized tests. Subsequently, an experimental phase is conducted to document both qualitative and quantitative data in real time while observing the interaction in simulated and real-world scenarios. Finally, the assessment gathers data over time by evaluating the system's adaptability throughout various phases of the disease via performance metrics including response times and error count as well as questionnaires.

2.3.1. Calibration Stage

Acquire user-specific parameters in order to adapt and customize the AAC system. The next four steps are: Conduct an initial session with each participant. Capture pertinent biometric information, including eye movements, motor abilities, and interaction preferences. Determine the precision and velocity of system interaction by conducting specific tests. Adapt the AAC system according to calibration results to ensure a customized experience.

2.3.2. Experimental Stage

To observe the interaction of the participants with the system in simulated and real scenarios. The next four steps are: Design simulated situations that represent common communication contexts. Introduce controlled scenarios (interaction modalities: mechanical, tactile, and visual) to evaluate the effectiveness and efficiency of the system. Record the interaction in real time, capturing quantitative data (response times, number of errors) and qualitative data (user comments, facial expressions). Provide immediate feedback to participants to adjust the system as needed.

2.3.3. Assessment Stage

Collect data on system adaptability as disease stages progress. The next five steps are: Perform regular assessments over time, reflecting different stages of the disease. Implement specific measures for each phase of the disease identified in Phase 2. Use questionnaires to assess the adaptability of the system at each stage. Document and analyze changes in user interaction and performance as the disease progresses. Integrate feedback from participants to make continuous adjustments and improve the adaptability of the system.

*2.4. Virtual Keyboard Assessment*

The AAC system implementation performed a sequential scan from left to right at regular intervals, identifying user interaction events with the characters displayed on the screen. These events were detected using three interaction modalities: mechanical, tactile, and visual. The metrics used were response times and number of errors detected during the interaction, complemented with a questionnaire for qualitative evaluation.

*2.5. State of the Disease Classification*

In order to address the categorization of neuromuscular disease states, we implemented fuzzy logic across three levels: initial, middle, and advanced. The primary objective was to anticipate the progression of the disease so that the AAC system could be modified in accordance with the particular stage or condition of the disease. This is achieved by applying triangular functions to the linguistic variables "initial", "middle", and "advanced". This approach leverages the expertise of movement disorder-specialized neurologists in order to delineate the stages of progression.

### 2.6. Performance Measures

For each writing test, time metrics and error counts were documented, in addition to a questionnaire utilized for qualitative assessment. The examiner was an expert on usability. Utilizing the heuristic method suggested by [33], the expert proceeded. Every metric (including satisfaction, general usability, efficiency, efficacy, and ease of learning) was evaluated in accordance with predetermined criteria and standards. The effectiveness and velocity with which users can complete the writing task were assessed in terms of efficiency. The effectiveness of the software was assessed by examining its capacity to produce the desired results without encountering any errors. The error rate was calculated by number of errors $\times$ 100)/total number of characters. Ease of learning: evaluated the speed with which users can learn to use the software, considering the user's opinion and intuitiveness. Satisfaction: evaluated the opinion of satisfaction when interacting with the software, through a simple survey. Usability: considered the overall user experience, integrating the previous metrics to obtain an overall view of the usability of the software. These evaluations were carried out systematically and with the participation of three real users to obtain practical feedback. In addition, the experts compared the results with standards in the field of usability and user experience.

### 2.7. Ethical Considerations

Within the framework of the current research, a university scientific ethics committee approved and supervised this study. This measure was taken to guarantee the implementation of fundamental ethical principles, including the promotion of equity in participant selection and the safeguarding of self-determination. Additionally, confidentiality and privacy of personal data were guaranteed, in accordance with established ethical standards, while protecting the rights and welfare of the individuals involved, through consent protocols and robust security measures.

## 3. Results

The AAC system was implemented using the C# programming language. Its design was inspired by a classic mobile phone keyboard and is based on an indirect access approach called "scanning". In this system, a periodic left-to-right scanning is performed at regular intervals. The user must generate an event when the system reaches the group of characters where the desired character is located. Similarly, another event must be produced when the system reaches the specific desired character.

Detection of these events is achieved through three distinct interaction methods: mechanical, tactile, and visual (see Table 1). The choice of the appropriate method is determined through an adaptation analysis. Additionally, the time intervals between the scans range from 1 s to 12 s (see Figure 2).

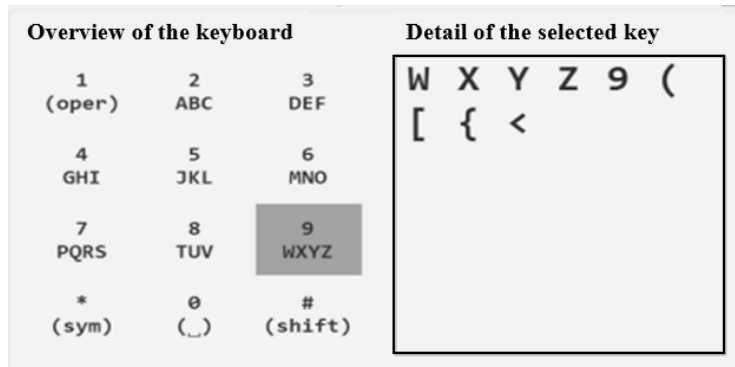

**Figure 2.** AAC system interface. Source: Authors.

**Table 1.** Interaction methods table. Source: Authors.

| Visual Method | Mechanical Method | Touch Method |
|---|---|---|
| Visual perception approaches, including gaze observation, eye movement tracking, and head motion-based indication technologies, have been extensively documented in academic literature [34]. Technological solutions based on gaze tracking are premised on monitoring a user's eye movements to infer the direction of their gaze [35]. In the specific context of AAC, non-invasive methods of eye tracking are presented as the most relevant option to address the everyday demands of users with motor skill limitations. | Mechanical and electromechanical devices used in AAC have applications in both direct and indirect access methods. In direct selection methods, users are presented with sets of options, requiring them to manually choose desired messages through voluntary input. This typically involves coordinating voluntary controls utilizing a specific body part, such as hands or fingers, or even a pointing device, to select a particular message [36]. | With the constant evolution of touchscreen technology, AAC applications incorporating touch activation have become a common presence in AAC direct selection systems. Several types of touchscreen technologies can be identified, such as resistive, capacitive, surface acoustic wave, and optical/infrared. Specifically, resistive and capacitive touchscreens are predominantly used in smart devices [19]. Resistive touchscreens operate by applying force or pressure from the user's fingers, whereas capacitive touchscreens are activated by the electrical charge present in the user's finger. |

The classification of stages in the progression of neuromuscular diseases becomes complex due to the variability in symptoms over time between two individuals affected by the same disease. In practical terms, determining the evolution of stages over time through conventional approaches entails substantial difficulties.

In this context, fuzzy logic emerges as an appropriate approach to address the classification of stages in neuromuscular diseases. A use case for fuzzy logic arises when it becomes necessary to classify imprecise phenomena based on expert knowledge and experience [37]. In this scenario, linguistic variables and their interpretations are provided by a neurologist specialized in movement disorders. The stages of progression in neuromuscular diseases are defined using three triangular functions corresponding to the linguistic variables "initial", "middle", and "advanced" (see Figure 3).

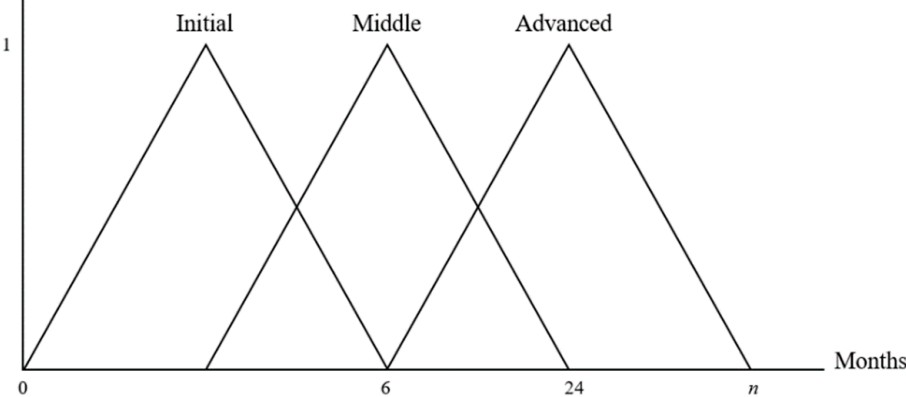

**Figure 3.** Fuzzy logic function for stages of neuromuscular diseases. Source: Authors.

The membership functions are given based on the linguistic variables defined as follows:

$$M(initial) = triangular(x, 6, 3) \tag{1}$$

$$M(middle) = triangular(x, 9, 12) \tag{2}$$

$$M(advanced) = triangular(x, 15, 24) \tag{3}$$

The fundamental realm is defined by an average lifetime reaching the precise value of (*n*). In this context, (*n*) symbolizes the uppermost age since the disease's onset, a value that notably differs among individuals.

Three overlapping regions inside the fuzzy sets give rise to two fuzzy sets. The important and difficult thing to determine is where these sets intersect. The diagnosis is done based on the symptoms and under the direction of a professional, preferably a neurologist who specializes in movement disorders.

### 3.1. Heuristic Rules for Determining Disease Stage in the System

When starting the system, the users are required to complete a set of four tests, conducted only once, aimed at classifying their disease stage (refer to Table 2). This series of tests allows to establish the optimal combination between the interaction method and the timing of the event.

**Table 2.** Captions for tables should be placed above the tables. Source: Authors.

| Tasks | Task Description |
| --- | --- |
| Task 1: | Press the "H" key |
| Task 2: | Perform a tap in the upper-left corner of a touchscreen. |
| Task 3: | Produce a sound with the mouth (snap). |
| Task 4: | Perform two winks within a one-second interval. |

The user is evaluated based on two main metrics:

Execution time: The time the user takes to complete a specific task designated by the system.

User error: Referring to one or several mistakes made unintentionally by the user during their interaction with the system.

The system's inference process was devised through a conditional structure, which utilizes the aforementioned metrics to determine the fuzzy set that best fits the user's characteristics (see Table 3). The four tests conducted are averaged based on these independent metrics. In situations where the metrics do not align across the four tests, the intermediate linguistic variable is employed to make a decision (see Table 4).

**Table 3.** Metrics for system inference. Source: Authors.

| Errors | Time Interval | Linguistic Variable |
| --- | --- | --- |
| 0–1 | 0.5 s–1 s | Initial |
| 2–3 | 1.1 s–2 s | Initial and middle |
| 3–5 | 2.1 s–4 s | Middle |
| 6–8 | 4.1 s–6 s | Middle and advanced |
| 9–12 | 6.1 s–10 s | Advanced |

**Table 4.** Interaction methods for fuzzy sets. Source: Authors.

| Stage | Interaction Methods |
|---|---|
| Initial | Mechanical, touch |
| | Touch |
| Initial and middle | Mechanical, touch |
| | Touch |
| Middle | Mechanical, touch |
| | Touch |
| | Visual |
| | Touch |
| Middle and advanced | Touch |
| | EGG |
| | Visual |
| Advanced | EGG |
| | Visual |

Fuzzy sets are specified in Table 4.

To project these rules into the form of fuzzy logic equations, we will use triangular membership functions for each of the stages (initial, middle, and advanced). The membership functions have been previously: Equations (1)–(3).

Where "*x*" represents the variable corresponding to the value of error or time.

Now, we can write the fuzzy logic equations for each of the stages based on the provided metrics:

3.1.1. Stage "*Initial*"

$$Initial(x) = min(M(initial)(x),\ M(initial)(y)) \tag{4}$$

where "*x*" represents the error and "*y*" represents the time. *min(M(initial)(x)* and *M(initial)(y)* are the triangular membership functions for the "*Initial*" stage based on the error and time ranges provided in Table 4.

3.1.2. Stage "*Initial and Middle*"

$$Initial\_Middle(x) = min(M(initial)(x),\ M(middle)(y)) \tag{5}$$

where "*x*" represents the error and "*y*" represents the time. *min(M(initial)(x)* and *M(middle)(y)* are the triangular membership functions for the "*Initial*" and "*Middle*" stages, respectively.

3.1.3. Stage "*Middle*"

$$Middle(x) = min(M(middle)(x),\ M(middle)(y)) \tag{6}$$

where "*x*" represents the error and "*y*" represents the time. *min(M(middle)(x)* and *M(middle)(y)* are the triangular membership functions for the "*Middle*" stage based on the error and time ranges provided in Table 4.

3.1.4. Stage "*Middle and Advanced*"

$$Middle\_Advanced(x) = min(M(middle)(x),\ M(advanced)(y)) \tag{7}$$

where "*x*" represents the error and "*y*" represents the time. *min(M(middle)(x)* and *M(advanced)(y)* are the triangular membership functions for the "Middle" and "Advanced" stages, respectively.

### 3.1.5. Stage "*Advanced*"

$$Advanced(x) = min(M(advanced)(x), \, M(advanced)(y)) \tag{8}$$

where "$x$" represents the error and "$y$" represents the time. $min(M(advanced)(x)$ and $M(advanced)(y)$ are the triangular membership functions for the "*Advanced*" stage based on the error and time ranges provided in Table 4.

The timeframe for categorizing the user under the linguistic variable remains open until they attain the "advanced" status, aligning with the interaction method recommended by the system. It is important to note that users have the option to disable the adaptability system at their discretion.

### 3.2. Machine Learning in ACC

In addition to the Fuzzy System, machine-learning techniques were integrated into the virtual keyboard, which in contrast to a conventional static design- evolves and adapts iteratively. This keyboard is characterized by the implementation of a machine learning-based system that constantly monitors and analyzes the lexical selections made by the user during communication. As the user interacts with the keyboard, the system records word preferences and their frequencies of use.

The essence of the process lies in the analysis stage, where machine learning comes into play. Through highly complex algorithms, the system identifies intrinsic patterns in lexical choices and groups them into semantically coherent categories. As these categories emerge, the keyboard begins to adapt in real time. Selecting a word or phrase becomes an extremely fluid act, as the keyboard anticipates and suggests the most plausible words based on the communicative context.

However, the uniqueness of this approach is further accentuated. The keyboard is not limited solely to contextual word prediction. Additionally, it generates a new row of words at the top of the keyboard, which houses the words most frequently used by the user. This personalized row is a dynamic entity, altering in accordance with the user's continuous interaction with the keyboard.

Consider a user who has a preference for words that are particularly relevant to their line of work or personal interests. As the user persists in interacting with the keyboard, these specific words gain priority and conveniently emerge in the new personalized row. This not only boosts communicative efficiency but also incites a feeling of empowerment and control in the user, as they observe how the keyboard adjusts to their idiosyncratic requirements.

## 4. Discussion

In this section, three case studies and usability assessment will be presented out of the ten included in the trial. For confidentiality reasons, the names have been altered. Additionally, before conducting the study, participants were asked to read and sign an informed consent form. Followed by a usability evaluation of the proposed software. In retrospect to this process, when analyzing the overall results of the case studies presented, a common pattern emerges of significant improvement in efficiency and communication satisfaction for the participants. The AAC system's adaptability was crucial, as it successfully adapted to the evolving requirements of users throughout different stages of the disease.

This final consideration emphasizes the importance of prior identification of the disease state of every participant, thereby highlighting the specific nature of the support tools utilized in each case. The initial scenario involved the utilization of an eye tracker, which emphasized the visual sensory modality. A speech generator was implemented as an assistive device in the second instance, whereas a head-mounted pointing device was employed in the third instance. Consequently, observations were carried out in adherence to these predetermined parameters, in order to assess the responsiveness of fuzzy logic components to the evolution of each pathological entity in conjunction with a variety of features.

### 4.1. Case Study 1: Study_subject_1—ALS Patient

Study_subject_1, a 45-year-old individual diagnosed with amyotrophic lateral sclerosis, volunteered to participate in the evaluation of our adaptive AAC system. As ALS progressed, Study_subject_1's ability to communicate verbally rapidly declined. We implemented our system on a tablet device equipped with eye-tracking technology, enabling Study_subject_1 to select symbols and phrases using eye movements.

Over a six-month period, we regularly assessed Study_subject_1's communication patterns and needs. The adaptive AAC system continuously analyzed his input, updating its predictive model to adapt to his changing motor abilities and communication preferences. As a result, the system provided Study_subject_1 with a personalized and dynamic communication interface.

The case study demonstrated that the adaptive AAC system successfully enhanced Study_subject_1's communication efficiency and overall satisfaction. Despite the progression of his condition, Study_subject_1 could maintain effective communication with his family, caregivers, and friends, accurately expressing his thoughts and emotions.

### 4.2. Case Study 2: Study_subject_2—Muscular Dystrophy Patient

Study_subject_2, a 32-year-old woman diagnosed with muscular dystrophy, took part in a longitudinal study involving the utilization of the adaptive AAC system. Study_subject_2's condition involved a gradual weakening of muscles and speech difficulties. To cater to her specific needs, we integrated the system into a voice-generating device accessible through switches.

The adaptive AAC system, employing a fuzzy logic-based approach, continuously learned from Study_subject_2's interactions and detected changes in her communication patterns. The system automatically adjusted its design and predictive capabilities to align with Study_subject_2's abilities at each stage of her condition.

Throughout the study, Study_subject_2 reported increased confidence in communication and a decrease in frustration compared to her prior experiences with non-adaptive AAC systems. The system's personalized nature allowed her to engage in conversations more effectively, preserving her autonomy and social interactions.

### 4.3. Case Study 3: Study_subject_3—Spinal Cord Injury Patient

Study_subject_3, a 28-year-old man with a spinal cord injury, participated in a home trial of the adaptive AAC system. His injury resulted in partial paralysis, affecting the motor functions of his upper limbs. To meet his needs, we integrated the system with a head-mounted pointer device.

During the trial, Study_subject_3 employed the adaptive AAC system in various real-life scenarios, such as communicating with family members, placing orders at restaurants, and engaging in leisure activities. The system's capacity to adapt to his motor abilities and preferences enabled Study_subject_3 to communicate effectively and independently across diverse environments.

The case study underscored how the adaptive AAC system positively impacted Study_subject_3's daily life, fostering social engagement and reducing barriers stemming from his physical condition.

The software evaluation followed the heuristic method proposed by [33]. The approach involves assessing usability based on four characteristics (efficiency, effectiveness, learnability, and satisfaction) and relying on experts to conduct the assessment. The current evaluation conducted on the proposed software yielded highly favourable outcomes, particularly in the following aspects: efficiency 90%; effectiveness 100%; learnability 90%; satisfaction 80%; overall usability 88%).

Proposed in the study [6] was a BCI system designed to assist individuals with severe motor disabilities with their communication abilities. The assessment included usability performance standards, subjective questionnaires, and observations on the interaction of the system with the virtual assistant. The results indicate a high degree of usability, exceeding the threshold set in advance. Nevertheless, concerns pertaining to the fluctuating

interpretations of voice commands and problems during interactions with the virtual assistant were recognized as obstacles. Furthermore, certain studies lacked a distinct correlation between the degree of disability and BCI performance, according to the research cited in [6].

A brain–computer interface (BCI) system was suggested by Martínez-Cagigal et al. for operating social networks on mobile phones. The system was evaluated in both healthy individuals and those who had motor disabilities. The findings demonstrate a precision rate of 92.3% among individuals in good health, and 80.6% among those with motor impairments. The evaluation encompassed both qualitative and quantitative metrics, such as output characters per minute (OCM), accuracy, and subjective questionnaires. The proposal is regarded as beneficial since it aims to increase individuals' independence and autonomy in their daily lives.

In our study, we present an AAC system that classifies stages of progression in neuromuscular diseases through the use of tactile, visual, and mechanical interaction methods with an emphasis on fuzzy logic. The intricacy of stage classification is demonstrated by the variability of symptoms exhibited by neuromuscular disease patients. Fuzzy logic is proposed as a potential solution for this fluctuation. Furthermore, the importance of incorporating visual, tactile, and mechanical perception methods into the system's implementation is emphasized.

Utilizing cutting-edge technologies including BCI and fuzzy logic, all three studies showcased substantial progress in the evolution of AAC systems. While research study 1 and 2 focus on communication via brain commands, the current research addressed staging in neuromuscular diseases. Qualitative assessment, which frequently employs subjective questionnaires for measuring usability and user satisfaction, is widely acknowledged. In light of the limitations identified in studies 1 and 2, including inconsistent interpretations of voice commands and difficulties in engaging with virtual assistants, the authors emphasized the need to enhance the systems' accuracy and adaptability. The latter can be enhanced by the research presented in this article, which analyzes its input continuously and updates its model to account for the user's evolving motor skills and communication preferences while pre-identifying the disease stage. On the other hand, the complexity in classifying disease stages in research study 3 highlights the need for further experimentation with fuzzy logic or other artificial intelligence methods to address symptom variability.

As per the literature review, Bircanin et al. [38] examine a number of challenges and opportunities associated with inherent characteristics of AAC systems, which are also relevant to the approach proposed in this research. In general terms, certain limitations are associated with the particular characteristics of each type of disability. For example, the research did not explicitly address individuals with significant sensory restrictions, constituting a gap in the comprehensive understanding of the system's capabilities Individuals who have hearing or vision impairments may encounter restrictions in the operation of systems that rely on these particular senses. An additional limitation that has been identified pertains to resistance to change, a phenomenon that is additionally influenced by the users' social circles and surroundings. The aforementioned resistance could potentially be observed in the degree of readiness with which certain patients adopt and adapt to the AAC system, thereby influencing their acceptability and performance. The thorough and individualized assessment of each patient is essential, as it enables the determination of the AAC system's relevance and permits adjustments that respond optimally to individual needs.

### 4.4. Limitations and Future Work

#### 4.4.1. Limitations

The effectiveness of the system is linked to the technology employed, which may rise to challenges concerning hardware limitations or technical malfunctions (e.g., processing capacity of the devices). However, it may require time and effort to customize the system

to the unique requirements of each user, and certain patients might experience a more pronounced learning curve.

On the other hand, the periodic scanning approach may not be fully customizable to suit individual user preferences. Adaptation is carried out through user-generated events, but this may be complex for those with very limited or too advanced motor skills. In addition, depending on the user's ability to generate events at specific times, there may be variability in the effectiveness of the system. Individual differences in the ability to generate events could affect the accuracy and speed of communication.

Additionally, the neurological specialists noted that individuals with the same disease might develop distinctive variations in symptoms or characteristics over time, which would pose an enormous barrier to staging. The system's capacity to accurately adapt to disease progression may be compromised by this factor. While fuzzy logic is employed to handle the staging of neuromuscular diseases, the inherent variability of symptoms can compromise classification accuracy, particularly in middle cases. Furthermore, the assessment of the three scenarios, which incorporates the utilization of visual, mechanical, and tactile methods for detecting events, could potentially impede accessibility for individuals with particular limitations on their capacity to employ said methods. There may be user populations for whom these methods are not optimal. In conclusion, with regard to technological democratization, the accessibility of specific users may be constrained by the device's functionality and technology.

### 4.4.2. Future Works

Several aspects emerge as possible lines of future research. The first is to explore the comparative efficacy of the support tools used (eye tracker, speech generator, pointer device) in a larger and more diverse sample of participants with different disease states. This could provide additional information on the adaptability and relative efficacy of each tool in different clinical settings. In general, to explore ways to improve the customization of the system to suit a broader spectrum of motor skills and individual preferences.

Furthermore, conducting a comparative analysis between the response of each support tool to disease evolution and the interaction between particular characteristics of fuzzy logic and machine learning algorithms would be of great value. By adopting this approach, a more profound comprehension of the fundamental mechanisms that contribute to the effectiveness of such tools in the management of various pathological states could be achieved.

Another line relates to the applicability of the system to other forms of motor disabilities and communication challenges. This raises the following inquiry: How could the system be adapted to address a more extensive spectrum of neuromuscular conditions? Subsequently, an investigation into alternative assistive systems designed for individuals afflicted with diverse neuromuscular disorders could be carried out. Furthermore, collaboration with neurologists who specialize in the field could take place to enhance and optimize the fuzzy logic functions, thereby guaranteeing a more accurate classification of the progression stages of neuromuscular diseases.

### 5. Conclusions

The key factor in the success of the adaptive AAC system was its customization capability. By continuously learning and adapting to changing motor skills and communication preferences of each individual, the system was able to meet the specific needs of each participant, maximizing their communication efficiency and overall satisfaction. The decision to utilize fuzzy logic was based on its inbuilt capability to accommodate the imprecision and uncertainty inherent in the data. Fuzzy logic is an ideal tool in the context of neuromuscular diseases, where stages of progression may lack accuracy. By capitalizing on the expertise and knowledge of professionals, this approach yields an adaptable, rule-driven depiction. Fuzzy logic demonstrated its suitability in capturing the inherent imprecision that characterizes the progression stages of these diseases.

Longitudinal studies proved to be beneficial in evaluating the effectiveness of the adaptive AAC system, as observed in the cases of Study_subject_1 and Study_subject_2. Assessing communication patterns and needs over time allowed researchers to observe the ongoing improvement of the system and gain a better understanding of its long-term impact on participants. The system's ability to integrate with various devices, such as eye-tracking technology, voice-generating devices with switch access, and head-mounted pointers, demonstrated its versatility. This adaptability enabled individuals with diverse motor disabilities to find a communication solution tailored to their specific needs.

The adaptive AAC system had a positive impact on the autonomy and social interactions of participants. By facilitating effective communication, it helped individuals engage in conversations with their families, caregivers, friends, and the community at large, thereby reducing frustration and promoting social participation. The system significantly contributed to enhancing the quality of life for individuals with motor disabilities, allowing them to express their thoughts and emotions accurately and to participate in real-life situations with greater independence and efficiency. While the studies focused on specific conditions (ALS, muscular dystrophy, and spinal cord injury), the positive results suggest that the adaptive AAC system could potentially be applied to other motor disabilities and communication challenges.

Case studies provide a strong foundation for future research and developments in the field of adaptive AAC systems. As technology advances and more user information is collected, continuous improvements can be made to enhance predictive capabilities and personalized communication features of the system. Furthermore, it should be evaluated whether the findings of this study can be extrapolated to other types of assistance systems for individuals with various forms of neuromuscular diseases.

Finally, the adaptive AAC system demonstrated promising results in enhancing communication efficiency and overall satisfaction for individuals with motor disabilities. Its ability to adapt and customize communication interfaces according to the needs of each user highlights its potential to positively impact the lives of many individuals facing communication challenges due to various conditions and disabilities. The synergy between machine learning and personalized adaptation in a virtual keyboard represents a significant milestone in communication technology aimed at individuals with motor limitations. This approach not only enhances communicative fluency and efficiency but also emphasizes the importance of autonomy and uniqueness in the expression process.

**Author Contributions:** Conceptualization, J.F.S.-Á., G.P.J.-Á. and J.A.J.-B.; methodology, J.F.S.-Á., G.P.J.-Á. and J.A.J.-B.; software, J.F.S.-Á. and G.P.J.-Á.; validation, J.F.S.-Á., G.P.J.-Á. and J.A.J.-B.; formal analysis, J.F.S.-Á., G.P.J.-Á. and J.A.J.-B.; investigation, J.F.S.-Á., G.P.J.-Á. and J.A.J.-B.; resources, J.F.S.-Á., G.P.J.-Á. and J.A.J.-B.; data curation, J.F.S.-Á. and G.P.J.-Á.; writing—original draft preparation, J.F.S.-Á., G.P.J.-Á. and J.A.J.-B.; writing—review and editing, J.F.S.-Á., G.P.J.-Á. and J.A.J.-B.; visualization, J.F.S.-Á. and J.A.J.-B.; supervision, G.P.J.-Á. and J.A.J.-B.; project administration, J.A.J.-B.; funding acquisition, J.F.S.-Á., G.P.J.-Á. and J.A.J.-B. All authors have read and agreed to the published version of the manuscript.

**Funding:** The APC was funded by the authors.

**Institutional Review Board Statement:** Ethic Committee Name: Comité de Bioética, Dirección de Investigación y Extensión, Universidad Nacional de Colombia Sede Medellín. Approval Code: HERMES-19697. Approval Date: 17 April 2021.

**Informed Consent Statement:** Informed consent was obtained from all subjects involved in the study.

**Data Availability Statement:** The data presented in this study are available on request from the corresponding author. The data are not publicly available due to due to the privacy of the information provided by the population with neuromuscular diseases studied.

**Acknowledgments:** The authors thank their institutions of origin, namely: Universidad Nacional de Colombia, and the Universidad Nacional Abierta y a Distancia.

**Conflicts of Interest:** The authors declare no conflicts of interest.

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
