# Peer review of "Facilitating Communication in Neuromuscular Diseases: An Adaptive Approach with Fuzzy Logic and Machine Learning in Augmentative and Alternative Communication Systems"

_computers, doi:10.3390/computers13010010_

Round 1
Reviewer 1 Report
Comments and Suggestions for Authors
The manuscript proposes a Fuzzy Logic and Machine Learning method in Augmentative and Alternative Communication Systems focusing on facilitating communication in neuromuscular diseases.
Some remarks:
1- The introduction presents much information but lacks references;
2- Line 184: variables must be in italics;
3- many equations must be numbered;
4- The authors mention the use of ML models. However, this is a wide field of research. Since they are submitting the manuscript to COMPUTERS JOURNAL, I was expecting a focus on further computational a further presentation and discussion on the computational issues of the proposal;
5- The discussion presented is just a presentation of 3 case studies, but the authors must present the results and point out the possible reasons for that observation.
6- Line 345: What does it mean "evolutionary approach"?
7- There are no future works;
8- The authors are proposing an interesting tool, but they should understand this is a proposal. They used extreme statements, such as "This approach lays the groundwork for a future in which technology adapts to humanity, rather than the other way around". Take care with this kind of self-praise.
Author Response
Dear Reviewer The authors have made all the proposed observations.Best regards.

Reviewer 2 Report
Comments and Suggestions for Authors
The Augmentative and alternative communication (AAC) system is beneficial for those with neuromuscular conditions. The research uses the fuzzy logic to ascertain the disease phase (initial, middle, and advanced) by employing intuitive guidelines. The results about the three case studies demonstrated promising results in enhancing communication efficiency and overall satisfaction for individuals with motor disabilities. The research has important practical significance.
The questions are:
1. Are there any cases that the AAC system may be not work for some patients when you test the system ?
2. About Table 2, Table 3 and Table 4, are "Captions for tables should be placed above the tables." and "Table captions should be placed above the tables." wrong ?
3. Are there any related works that combine the machine learning technique with ACC?
4. What are the disadvantages of your proposed?
5. What is the reason that you choose the Fuzzy Logic method? In the experiments, have you tried other machine learning methods?
Author Response
Respected editors.The authors made all the proposed changes.
Please review the attached file, which indicates the changes made.
Best regards,

Reviewer 3 Report
Comments and Suggestions for Authors
The article "Facilitating Communication in Neuromuscular Diseases: An Adaptive Approach with Fuzzy Logic and Machine Learning in Augmentative and Alternative Communication Systems" where they proposes the use of the technique based on fuzzy logic to determine the phase of the disease through the use of intuitive guidelines. While the research topic is significant and timely, there are several areas where the article could benefit from improvements.
1. Lack of Clarity in Problem Statement:
The article fails to clearly articulate the specific problem or gap that the proposed algorithm seeks to address. It would greatly enhance the article's quality and reader understanding if the authors explicitly state the limitations of algorithms and founding of the proposed technique, providing a clear rationale for the need for applied both algorithms.
2. Insufficient Methodological Detail:
The article lacks comprehensive methodological detail, making it difficult for readers to replicate or validate the proposed algorithm. It is crucial to provide a step-by-step description of the algorithm, including specific equations, formulas, or mathematical operations. This will enable researchers to evaluate its effectiveness and potentially build upon it in future studies.
3. Inadequate Evaluation and Comparison:
The authors briefly mention the performance evaluation of their algorithm but fail to provide a comprehensive analysis. It is essential to compare the proposed algorithm's results with existing state-of-the-art methods, employing appropriate evaluation metrics such as accuracy, precision, recall, and score. This will allow readers to understand the algorithm's efficacy and performance in real-world scenarios.
4. Lack of Discussion on Limitations:
The article lacks discussion on the limitations and potential challenges that might affect the proposed algorithm's practical implementation. Addressing these limitations and providing insights into how they could be mitigated or improved upon will enhance the article's overall value and applicability.
5. Inadequate Experimental Setup:
The experimental setup provided in the article lacks necessary details. Information regarding the dataset used, sample size, diversity of subjects, and any pre-processing techniques employed is crucial. Without this information, it is challenging to gauge how representative the experimental results are and whether they can be generalized to other real-world scenarios.
6. Insufficient Analysis of Results:
The article lacks a thorough analysis of the obtained results. The authors should interpret and discuss the findings in detail, outlining any patterns, trends, or anomalies observed. It would be beneficial to provide visualizations, charts, or graphs to assist readers in comprehending the results.
7. Lack of Ethical Considerations:
As AAS systems uses information of test persons, it is important to discuss the ethical considerations associated with their usage. The article should address issues related to data privacy, informed consent, and compliance with ethical guidelines.
8. Lack of abstract:
The summary presented by the authors must be improved and rewritten, because it makes no sense and the sentences are disjointed. A conscious analysis is not carried out on the technique presented, likewise, it is not mentioned what it contributes to the field of study and how these techniques presented together are better than others (they require a comparison of performance)
9. Lack of the conclusions:
Overall, the article "Facilitating Communication in Neuromuscular Diseases: An Adaptive Approach with Fuzzy Logic and Machine Learning in Augmentative and Alternative Communication Systems" possesses significant potential but not definitive. Incorporating the aforementioned suggestions, such as improving the problem statement, enhancing methodological detail, thorough evaluation, accounting for limitations, refining the experimental setup, analyzing results in-depth, and considering ethical considerations, would considerably strengthen the article's impact and credibility in the field.
Comments on the Quality of English Language Before analyzing the quality of the English level,it is necessary to make fundamental adjustments in general
to the entire document. Yes, there are errors in the text,
but the main need is to substantiate the proposal
Author Response
Respected reviewer.The authors made all the proposed changes.
Please review the attached file, which indicates the changes made.
Best regard,

Round 2
Reviewer 1 Report
Comments and Suggestions for Authors
Accept
Reviewer 2 Report
Comments and Suggestions for Authors
Accepted.